# Unavoidable Destroyed Exergy in Crude Oil Pipelines due to Wax Precipitation

**DOI:** 10.3390/e21010058

**Published:** 2019-01-12

**Authors:** Qinglin Cheng, JinWei Yang, Anbo Zheng, Lu Yang, Yifan Gan, Yang Liu

**Affiliations:** Key Lab of Ministry of Education for Enhancing the Oil and Gas Recovery Ratio, Northeast Petroleum University, Daqing 163318, China

**Keywords:** waxy crude oil, pipeline transportation, characteristic temperature, the unavoidable destroyed exergy

## Abstract

Based on the technological requirements related to waxy crude oil pipeline transportation, both unavoidable and avoidable destroyed exergy are defined. Considering the changing characteristics of flow pattern and flow regime over the course of the oil transportation process, a method of dividing station oil pipelines into transportation intervals is suggested according to characteristic temperatures, such as the wax precipitation point and abnormal point. The critical transition temperature and the specific heat capacity of waxy crude oil are calculated, and an unavoidable destroyed exergy formula is derived. Then, taking the Daqing oil pipeline as an example, unavoidable destroyed exergy in various transportation intervals are calculated during the actual processes. Furthermore, the influential rules under various design and operation parameters are further analyzed. The maximum and minimum unavoidable destroyed exergy are 381.128 kJ/s and 30.259 kJ/s. When the design parameters are simulated, and the maximum unavoidable destroyed exergy is 625 kJ/s at the diameter about 250 mm. With the increase of insulation layer thickness, the unavoidable destroyed exergy decreases continuously, and the minimum unavoidable destroyed exergy is 22 kJ/s at 30 mm. And the burial depth has little effect on the unavoidable destroyed exergy. When the operation parameters are simulated, the destroyed exergy increases, but it is less affected by the outlet pressure. The increase amplitude of unavoidable destroyed exergy will not exceed 2% after the throughput rises to 80 m^3^/h. When the outlet temperature increases until 65 °C, the loss increase range will not exceed 4%. Thus, this study provides a theoretical basis for the safe and economical transportation of waxy crude oil.

## 1. Introduction

The crude oil produced in China is mostly condensate and viscous crude oil. Heated transportation is often used in pipe transmission, which often leads to high energy consumption in pipeline operations. Building a scientific evaluation system for energy consumption is important for energy efficiency management [1]. For example, Wang [2] introduced the ton oil gas consumption coefficient *φ* and suggested a new method of gas consumption evaluation for crude oil gathering and transportation systems. The advantages and disadvantages of the management process can be evaluated by comparing *φ* to determine the disparity between various systems and oil fields and promote the development of energy-saving work. Zuo et al. [3] referred to the concept of hydraulic horse power (HHP)—used by Enbridge and other pipeline companies—and introduced two new energy consumption evaluation indices for long-distance pipelines. These indices shift the discussion of energy consumption from the power station (oil station/compressor station) to the energy consuming section itself. Probert et al. [4] provided a theoretical basis for predicting the optimal thickness of insulation materials required for such pipelines, so as to achieve the smallest energy operation. Nguyen et al. [5] show that the oil platform is an energy-intensive system. Referring to the nature of oil, export specifications and oil field life, the energy used by each facility ranges from several megawatts to several hundred megawatts, and the overall system is energy-saving in terms from equipment size and process integration.

In recent years, exergy has been regarded as an energy consumption evaluation index in thermodynamics. Exergy denotes that part of energy quantity that can be transformed into useful work to the utmost degree when substance or material flow changes reversibly from any state to a given environment equilibrium state. The concept exergy, which organically combines “quality” and “quantity” of energy and reflects the real value of energy, has made all kinds of energy comparable and solved the puzzling problem that none of the parameters can be singly used to assess the value of energy in thermodynamics and energy science. It also changed traditional ideas about energy character, energy loss and energy transformation efficiency, etc [6,7,8,9].

The exergy analysis method addresses the deficiency of the first law of thermodynamics, which simply analyzes energy from the “quantity” relation and has been introduced into the energy consumption analysis of crude oil pipelines in recent years. Zhang [10] stated that the crude oil transportation process must be accompanied by the consumption of propulsion exergy. Based on this exergy analysis principle, the exergy analytical model of crude oil transportation is established. The formula for propulsion exergy consumption is derived, and the change in propulsion exergy consumption due to operation parameters is analyzed to lay the foundation for pipe exergy flow classification. Based on external destroyed exergy and internal exergy dissipation, Li [11], starting with the substance being transported in the pipeline, published a study dividing exergy flow into effective exergy consumption and invalid exergy consumption and derived a method to calculate them.

Conventional exergy analysis can only suggest the potential or possibility of thermodynamic process improvements, but it cannot note whether the improvement is feasible. In the actual thermodynamic process, temperature difference, pressure difference and chemical potential difference are driving forces. The existence of driving forces inevitably leads to destroyed exergy—and the greater the driving force, the faster the process, and the greater the destroyed exergy. However, to carry out the thermal process and cycle, there must be a driving force, which inevitably leads to destroyed exergy [12].

Therefore, for pipeline transportation with irreversible phenomena, such as temperature difference and friction flow, effective improvement measures based on differentiating unavoidable and avoidable destroyed exergy are necessary for energy savings. This analysis method accurately points out the maximum destroyed exergy location, can provide a basis for identifying the weak links of energy utilization and can be used to analyze the imperfection of various thermodynamic cycles. The application of thermodynamics in engineering practice has been broadened. It can be concluded that the method can provide accurate, reliable and valuable research and analysis results for various industrial sectors.

## 2. Definition of Unavoidable Destroyed Exergy

Tsatsaronis and Park et al. first brought up the concept of unavoidable destroyed exergy in their paper in 2002 [13]. For crude oil transportation, there must be a theoretical temperature drop and theoretical pressure drop to ensure the safe and economical transportation of crude oil; the corresponding destroyed exergy is unavoidable destroyed exergy. The excess theoretical value in the actual transportation process is avoidable destroyed exergy, i.e., in the actual conveying process, the part that exceeds the theoretical value is avoidable destroyed exergy. In this way, the destroyed exergy of conventional exergy analysis can be divided into two parts: unavoidable destroyed exergy $E_{x,U}$ and avoidable destroyed exergy $E_{x,A}$; thus, total destroyed exergy $E_{x}$ is:(1)$$E_{x}=E_{x,U}+E_{x,A}$$

The heat loss and friction loss in pipe transportation are considered to correspond to thermal destroyed exergy and pressure destroyed exergy, respectively. There must be a technical or economic theoretical minimum temperature difference in the crude oil transportation process, and the thermal destroyed exergy caused by corresponding temperature difference Δ*T* is the minimum thermal destroyed exergy. The driving force used to overcome oil flow viscous frictional resistance and ensure the smooth advance of the crude oil from the initial point to the end of the pipe is the minimum pressure destroyed exergy of the start and end pressure drop Δ*p*. Therefore, to ensure safe and economical operation of crude oil pipelines, the theoretical temperature and pressure drops are obtained, so there exists corresponding theoretical minimums $E_{x}{\left(T\right)}_{\min}$ and $E_{x}{\left(p\right)}_{\min}$ of thermal destroyed exergy and pressure destroyed exergy. Then, their sum is the unavoidable destroyed exergy in the crude oil transportation process [12], thus: (2)$$E_{x,U}=E_{x}{\left(T\right)}_{\min}+E_{x}{\left(p\right)}_{\min}$$

After dividing destroyed exergy into avoidable and unavoidable destroyed exergy, the improvement of the crude oil pipeline process will change from a general reduction of destroyed exergy to the decrease of avoidable destroyed exergy and minimization of unavoidable destroyed exergy.

## 3. Determination of Unavoidable Destroyed Exergy for Waxy Crude Oil

With the decrease in the pipeline temperature and the precipitation of the wax crystals in the oil, the crude oil may be transformed from a Newtonian fluid to a non-Newtonian fluid. With the reduction of pipeline transportation throughput and the decrease of the pipeline transport temperature, the tube segments of the non-Newtonian flow pattern will lengthen [14]. Considering the different rheological properties of crude oil in different temperature ranges, a method is proposed to divide the oil pipeline into transportation intervals according to characteristic temperatures, such as wax appearance point, critical transition temperature and anomalous point. A formula for calculating the unavoidable destroyed exergy of crude oil under different flow conditions is derived. 

### 3.1. Crude Oil Critical Transition Temperature

Considering the stable operation of an oil pipeline, the Reynolds number of the same crude oil can be expressed as a function of temperature. The critical transformation temperature of the flow pattern and flow regime can be calculated by the critical Reynolds number inverse calculation. In this method, traditional flow regime judgment (based on Reynolds number) and flow pattern judgment (using the anomalous point $T_{F}$ as the standard) are unified to use the pipeline temperature as the basis for judgment [15,16].

For Newtonian waxy crude oil, the critical Reynolds number is 2000 for internal flow [17] when the flow of the Newtonian fluid is converted from laminar to turbulent. Reynolds number is defined as:(3)Re=Dvυ=Dρvμ
where, D—Inner diameter of oil pipeline, m; v—Crude oil flow velocity, m/s; υ—Oil flow viscosity, m^2^/s.

It should be noted that when calculating the transition temperature, the viscosity of the oil flow for Newtonian flow is calculated based on the critical Reynolds number Rec. According to its viscosity temperature curve, $T_{c}$ is obtained.

For non-Newtonian waxy crude oil, the specification provides that the critical Reynolds number is 2000, which marks the change from laminar flow to turbulent flow, and the Reynolds number is defined as follows [17]:(4)ReMRc=ρDn′v2−n′K′8(6n′+2n′)n′
where, ρ—Density of crude oil, kg/m^3^; K′—Consistency coefficient of crude oil, Pa·s; n′—Rheological index of crude oil.

When waxy crude oil becomes a non-Newtonian fluid, it is usually a pseudoplastic fluid. In the temperature range of pseudoplastic fluid, the definition ReMRc and its rheological parameters n′ and K′ can be expressed as a function of temperature. Using Formulas (5) and (6), the flow transition temperature can be back calculated. That is [18]:(5)$$K\prime =A_{1}e^{-B_{1}T}$$
(6)$$n\prime =A_{2}+B_{2}T$$
where, $A_{1}$, $A_{2}$, $B_{1}$, $B_{2}$—constant.

### 3.2. Determination of the Specific Heat Capacity of Waxy Crude Oil

It should be noted that wax deposition may occur during the transportation of waxy crude oil. The total heat transfer coefficient of the pipeline gradually decreases with the increase of the thickness of the wax layer. There will be a wax-free section, an initial paraffin section and a wax deposition tail section in the oil pipeline, and the specific heat capacities of the oil are different in each section:

(1) When the oil temperature is higher than the wax appearance temperature of waxy crude oil, there is no wax phenomenon in the crude oil pipeline (this is called the wax-free section). In this temperature range, the specific heat capacity of crude oil increases slowly with the rise of temperature. The relationship is [19]:(7)$$c=\frac{1}{\sqrt{\rho_{4}^{15}}}(1.687+3.39\times 10^{-3}T)$$
where, c—The crude oil specific heat capacity, kJ/(kg·°C); $\rho_{4}^{15}$—The crude oil relative density at 15 °C, dimensionless; *T*— the crude oil temperature, °C.

(2) When the temperature of the oil is lower than the wax appearance point, the wax will precipitate. In particular, after the oil temperature drops to the wax appearance point of crude oil, the wax in the pipeline will increase, and the wax layer will gradually thicken along the oil delivery direction until a peak is reached—that is, the maximum thickness of wax deposition. This section is called the initial paraffin section. Due to the release heat from the wax crystal in the above temperature range, the specific heat capacity of crude oil increases with decreasing temperature, and the relationship between crude oil and temperature fits the following formula [19]:(8)$$c=4.186-Ae^{aT}$$
where, A—constant, different from crude oil, kJ/(kg·°C); *a*—constant, different from crude oil, 1/°C.

(3) From the peak point of wax deposition to the interval of crude oil intakes, it is called the wax deposition tail section. With the increase of wax deposit thickness, the operation conditions of oil pipelines correspondingly change. In this temperature range, the specific heat capacity of crude oil decreases with decreasing oil temperature. The relationship between specific heat and crude oil temperature in this interval can be expressed as [19]:(9)$$c=4.186-Be^{-mT}$$
where, *B*—constant, different from crude oil, kJ/(kg·°C); *m*—constant, different from crude oil, 1/°C.

The parameters of several waxy crude oils are shown in Table 1.

### 3.3. Temperature Range of Pipeline Transportation

In the process of pipeline transportation, the specific heat capacity of oil and viscous frictional resistance vary with temperature; thus, the oil pipeline is divided into temperature intervals according to the wax appearance point $T_{sl}$, anomalous point $T_{F}$ and flow transition temperature $T_{c}$, $T_{MRc}$. Specifically [20,21,22]:

(1) If the temperature range is $T_{z}\geq T_{sl}$, there is only a single temperature range $T_{R}\geq T\geq T_{z}$ in the pipeline transportation, and the crude oil will remain in the Newtonian turbulence state;

(2) If the temperature range is $T_{sl}\geq T_{z}$ and $T_{z}\geq T_{c}$, the pipeline transportation process is divided into two temperature intervals $T_{R}\geq T\geq T_{sl}$ and $T_{sl}\geq T\geq T_{z}$; thus, the pipeline crude oil is in the Newtonian flow state;

(3) If the temperature range is $T_{sl}\geq T_{z}$ and $T_{z}\geq T_{F}$, the pipeline transportation process is divided into three temperature intervals $T_{R}\geq T\geq T_{sl}$, $T_{sl}\geq T\geq T_{c}$ and $T_{c}\geq T\geq T_{z}$; thus, the pipeline crude oil is in Newtonian flow and there may be laminar flow state;

(4) If the temperature range is $T_{sl}\geq T_{z}$ and $T_{z}\geq T_{MRc}$, pipeline transportation is divided into four temperature intervals $T_{R}\geq T\geq T_{sl}$, $T_{sl}\geq T\geq T_{c}$, $T_{c}\geq T\geq T_{F}$ and $T_{F}\geq T\geq T_{z}$; thus, non-Newtonian fluid characteristics begin to appear;

(5) If the temperature range is $T_{sl}\geq T_{z}$ and $T_{z}\leq T_{MRc}$, pipeline transportation is divided into five temperature intervals $T_{R}\geq T\geq T_{sl}$, $T_{sl}\geq T\geq T_{c}$, $T_{c}\geq T\geq T_{F}$, $T_{F}\geq T\geq T_{MRc}$ and $T_{MRc}\geq T\geq T_{z}$; thus, the pipeline crude oil presents non-Newtonian laminar flow in some pipe sections.

The division of the temperature range for a waxy crude oil pipeline is shown in Figure 1.

During crude oil transportation, $T_{0}$ is the ambient temperature around the pipeline, *G* is the throughput of oil transported by the pipeline and *i* is the hydraulic gradient. When fluid flows through pipe section d*L*, the corresponding temperature drop is d*T*. The energy balance of elementary section d*L* under stable conditions for a waxy crude oil pipeline is [23]:(10)$$K\pi D(T-T_{0})dL=-GcdT+gGidL$$

For Formula (10), the left is the heat dissipation from the d*L* pipe to the surrounding medium in unit time. The first item at the right is the heat release from the temperature drop d*T* of the oil flow in the pipe. The second item is the heat transformed by friction loss of oil flow in d*L* section.

The total length of the wax oil pipeline is the sum of the temperature range lengths for each oil transportation, and the temperature intervals lengths of the pipeline are calculated. The length of each temperature range $L_{i}$ can be obtained according to the Formula (11).
(11)$$L_{i}=\frac{Gc}{K\pi D}\ln\frac{T_{R}-(T_{0}+b)}{T_{L}-(T_{0}+b)}\left(b=\frac{giG}{K\pi D}\right)$$
where, $T_{R}$—Initial point temperature of the crude oil pipeline transportation temperature range, °C; $T_{L}$—Terminal point temperature of the crude oil pipeline transportation temperature range, °C.

### 3.4. Calculation of Unavoidable Destroyed Exergy 

#### 3.4.1. Unavoidable Thermal Destroyed Exergy

According to the definition of unavoidable destroyed exergy, its calculation can be simplified as the calculation of the theoretical minimum temperature difference for pipe transmission. Considering the influence of leakage and insulation damage during the actual operation of a crude oil pipeline, the measured terminal temperature will generally be lower than the theoretical terminal oil temperature. Therefore, under the premise of meeting crude oil pipeline transportation technology requirements, the theoretical terminal temperature $T_{zl}$ is generally 3 °C above the condensation point. The theoretical minimum temperature difference ΔT is the difference between the theoretical initial point temperature of the oil pipeline and the theoretical terminal temperature, and the theoretical outstation temperature $T_{Rl}$ of the crude oil can be derived using the Sukhov formula [23]. That is:(12)$$T_{Rl}=T_{0}+(T_{Zl}-T_{0})e^{-aL}$$

Considering the different temperature ranges in crude oil transportation, it is necessary to deduce the theoretical initial and terminal temperatures in each temperature range from the theoretical end temperature. Then, the theoretical outstation station temperature is:(13)$$T_{Rl}=T_{0}+(T_{Zl,i}-T_{0})e^{-aL_{i}}\;\left(a=\frac{K\pi D}{Gc}\right)$$

The unavoidable thermal destroyed exergy in the crude oil transportation process is the sum of the unavoidable thermal destroyed exergy from each temperature range.
(14)$$E_{x,U}(T)=\sum_{i=1}^{n}Gc(T_{Ri}-T_{Li}-T_{0}\ln\frac{T_{Ri}}{T_{Li}})$$

#### 3.4.2. Unavoidable Pressure Destroyed Exergy

During crude oil transportation, when fluid flows through pipe section dl, the pressure difference corresponding to that pipe section is Δp. The stress area is the pipeline cross section, so the unavoidable pressure destroyed exergy in this section is:(15)$$E_{x,U}(p)=\Delta p\times SdL$$
where *S*—Pipe cross-section area, m^2^.

The unavoidable pressure destroyed exergy in the whole pipe section is:(16)$$E_{x,U}(p)=\int_{0}^{L}\Delta p\times SdL$$

The calculation of frictional pressure drop must consider the flow pattern and flow regime of crude oil during transportation. For Newtonian fluids, the formula for pipeline frictional pressure drop is:(17)$$\Delta p=\rho gh\;\left(h=\beta \frac{Q^{2-m}\nu_{p}^{m}}{D^{5-m}}L\right)$$
where, *h*—Friction loss, m; *Q*—Pipe volume flow, m^3^/s; *L*—Pipeline length, m.

For non-Newtonian fluid, the formula for pipeline frictional pressure drop is:

For laminar flow [15]:(18)$$\Delta p=\frac{4LK}{D}{\left(\frac{3n+1}{4n}\right)}^{n}{\left(\frac{8v}{D}\right)}^{n}=4LK{\left[\frac{8(3n+1)}{\pi n}\right]}^{n}\frac{Q^{n}}{D^{3n+1}}$$

For turbulent flow [15]:(19)$$\Delta p=\lambda \frac{L}{D}\frac{\rho v^{2}}{2}=4f\frac{L}{D}\frac{\rho v^{2}}{2}$$
where, λ, f—Friction coefficient of power law fluid, which generally uses the Dodge-Metzner semi-empirical formula for confirmation:(20)1f=4.0n′0.75lg[ReMRf(1−n′2)]−0.4n′1.2

Finally, when calculating the unavoidable pressure destroyed exergy, it is necessary to combine the pipe temperature range. The unavoidable destroyed exergy of each interval can be obtained using the integral of (18), wherein the temperature range of Newtonian crude is:(21)$$E_{x,Ui}(p)=\int \frac{\pi }{4}\beta \frac{Q^{2-m}\nu^{m}L_{i}\rho g}{D^{3-m}}dL=\frac{\pi \beta Q^{2-m}\nu^{m}{L_{i}}^{2}\rho g}{8D^{3-m}}$$

For the temperature range of non-Newtonian crude oil.

For laminar flow:(22)$$E_{x,Ui}(p)=\frac{\pi K\prime Q^{n\prime }{L_{i}}^{2}}{2D^{3n\prime -1}}{\left[\frac{8(3n\prime +1)}{\pi n\prime }\right]}^{n\prime }$$

For turbulent flow:(23)$$E_{x,Ui}(p)=\int \frac{\pi }{4}\times 0.3304f\frac{Q^{2}}{D^{5}}\times \rho gL_{i}dL=0.1297f\rho g\frac{Q^{2}{L_{i}}^{2}}{D^{5}}$$

Therefore, the unavoidable pressure destroyed exergy during crude oil transportation is the sum of the unavoidable pressure destroyed exergy from each temperature range:(24)$$E_{x,U}(p)=\sum_{i=1}^{n}E_{x,Ui}(p)$$

### 3.5. The Calculation Process of Unavoidable Destroyed Exergy

To summarize, the unavoidable destroyed exergy calculation process for pipeline transportation is shown in Figure 2.

## 4. Calculation Example

### 4.1. Basic Data

Taking an oil pipeline in the Daqing oilfield as an example, crude oil pipeline transportation is simulated; the basic operating and physical parameters of the crude oil pipeline are shown in Table 2.

#### 4.1.1. Critical Transition Temperature

Using the basic equations, Newtonian flow transition temperature $T_{c}$ is 30.8 °C, and non-Newtonian transformation temperature $T_{MRc}$ is 32.1 °C.

#### 4.1.2. Pipeline Transmission Temperature Intervals

According to the axial temperature distribution curve of the oil pipeline, the crude oil inlet station temperature is 36.2 °C. Therefore, the whole oil pipeline can be divided into three intervals according to the transportation temperature:

Newtonian hydraulic smooth region (above the wax appearance point, $T_{R}>T>T_{sl}$), Newtonian hydraulic smooth region (below the wax appearance point, $T_{sl}>T>T_{F}$) and Non-Newtonian turbulent region ($T_{F}>T>T_{Z}$); the interval lengths are 20,755 m, 18,950 m and 5414 m, respectively, and the temperature region of waxy crude oil is obtained as shown in Figure 3:

#### 4.1.3. Theoretical Outstation Temperature

The waxy crude oil condensation point is 25 °C and the theoretical inlet temperature $T_{lz}$ is 28 °C, which meet crude oil pipeline transportation technology requirements. According to the formula, the outstation temperatures of the crude oil in the Non-Newtonian turbulence interval, Newtonian turbulent wax precipitation interval and Newtonian turbulent non-wax precipitation interval are 38.54 °C, 45.28 °C and 46.05 °C, respectively.

### 4.2. Calculation Results and Analysis of Unavoidable Destroyed Exergy

According to the basic pipeline transportation data, the unavoidable destroyed exergy of the oil pipeline is calculated. The results are shown in Table 3.

According to the data in the table, the unavoidable thermal destroyed exergy in the pipeline transportation of wax crude oil accounts for an overwhelming majority of the unavoidable destroyed exergy (95–99%). This further proves the decisive role of temperature research in the analysis of crude oil transportation. By comparing the temperature regions of pipeline transmission, unavoidable destroyed exergy is larger in the interval $T_{R}>T>T_{sl}$. This is mainly due to the higher temperature of crude oil at the initial stage of pipeline transportation and the larger temperature difference and larger pressure difference between the surrounding environment and the medium. In the $T_{sl}>T>T_{F}$ interval, the ratio of unavoidable thermal destroyed exergy increases in the pipeline. This is mainly because, with the gradual decrease of pipe temperature, pipe heat dissipation correspondingly decreases relative to the last pipe transportation interval; thus, there is a great reduction of thermal destroyed exergy. Because of wax precipitation in pipeline, the resistance increases, so the pressure difference increases. Therefore the unavoidable pressure destroyed exergy rises rapidly.

In the interval $T_{F}>T>T_{Z}$, unavoidable destroyed exergy is smallest. The main reason for this is that the operation temperature and pressure of the pipeline are decreasing, and the trend is gentle in late-stage oil pipeline transportation. The state is close to that of the surrounding medium, and temperature and pressure differences decrease continuously, which minimizes the unavoidable destroyed exergy of the pipeline. 

According to the basic data of the pipeline transport process, unavoidable pressure destroyed exergy, unavoidable thermal destroyed exergy and unavoidable total destroyed exergy are calculated; their distribution along distance shown in Figure 4.

It can be seen from the figures that the curve of unavoidable destroyed exergy in Newtonian hydraulic smooth intervals ($T_{q}>T>T_{sl}$, $T_{sl}>T>T_{F}$) is greater than that of the non-Newtonian turbulence intervals, which considers the changing flow pattern and flow regime with the pipeline transport temperature.

For all Newtonian hydraulic smooth intervals, unavoidable thermal destroyed exergy has an approximately linear relationship with pipeline length. The wax layer is formed by the precipitation of wax crystals in the Newtonian hydraulic smooth wax precipitation interval, which has a thermal insulation effect; thus, the calculated value of unavoidable thermal destroyed exergy is the lowest. Moreover, the crude oil viscosity increases gradually with the decrease of the pipeline transportation temperature. Both the pressure drop for crude oil safety and economic transportation and the calculated value of unavoidable pressure destroyed exergy increase gradually.

### 4.3. Influence of Design Parameters on Unavoidable Destroyed Exergy Loss

The parameters used to analyze unavoidable destroyed exergy in pipeline transportation may be design parameters or operation parameters. Correspondingly, there are two types of analysis: design exergy analysis and operation exergy analysis. Design exergy analysis evaluates the design quality of the oil pipeline in terms of energy consumption to improve pipeline design and reduce energy consumption. This chapter studies unavoidable destroyed exergy in terms of oil pipe diameter, insulation thickness and buried depth.

#### 4.3.1. Change with Pipeline Diameter

Waxy crude oil diameter is varied (168, 219, 273 and 323 mm). Destroyed exergy of pipe transportation is calculated and destroyed exergy distribution curves are shown in Figure 5.

From the curves, it can be seen that flow velocity in the crude oil pipe decreases with the increase of pipeline diameter, which makes the pressure difference reduce gradually to ensure the safe and economical crude oil transportation. Correspondingly, unavoidable pressure destroyed exergy decreases gradually during pipeline transportation. However, because of its smaller proportion, unavoidable thermal destroyed exergy during pipeline transportation is basically same as unavoidable total destroyed exergy. That is, it first increases significantly, then decreases, but the overall variation does not exceed 15%. Due to the rise of flow, the Newtonian wax precipitation interval lengthens, and the Newtonian no-wax precipitation interval shortens. When the pipe diameter is 323 mm, Non-Newtonian turbulent flow occurs in the pipeline, and the overall heat transfer coefficient of the oil pipeline increases first and then decreases. Accordingly, the temperature difference first increases and then decreases as a driving force for crude oil transportation, which significantly increases the unavoidable thermal destroyed exergy. Additionally, the total destroyed exergy in pipe transportation increases with pipe diameter. Therefore, avoidable destroyed exergy first decreases and then increases with the changed pipe diameter. Thus, save energy and reduce consumption for a waxy crude oil pipeline, the optimal diameter must to be determined. The length and terminal temperature of the crude oil pipeline under different diameters are shown in Table 4:

#### 4.3.2. Change with Insulation Layer Thickness

Waxy crude insulation layer thickness is varied (20, 30, 40, 50 and 60 mm). Destroyed exergy during pipe transportation is calculated and destroyed exergy distribution curves are shown in Figure 6.

It can be seen from the curves in the figure that increasing insulation thickness first decrease pressure destroyed exergy and then gradually increases it. However, because of its small proportion, unavoidable thermal destroyed exergy in pipeline transportation is basically consistent with the downward trend of unavoidable total destroyed exergy. This is because, with the gradual increase of insulation layer thickness, the total heat transfer coefficient of the pipe gradually decreases, and the insulation performance improves. The temperature difference decreases correspondingly to ensure smooth transportation, which decreases the unavoidable destroyed exergy. Additionally, total destroyed exergy in the pipeline decreases with increased insulation layer thickness, and the reduction trend slows. Therefore, to save energy and optimize waxy crude oil pipeline operations, it is necessary to consider the relationship between insulation layer thickness and destroyed exergy to determine the optimum thickness of the insulation layer.

#### 4.3.3. Change with Buried Depth

The waxy crude buried depth is varied (800, 1200, 1600, 1800 and 2000 mm). Destroyed exergy during pipe transportation are calculated and destroyed exergy distribution curves are shown in Figure 7.

It can be seen in the curves of the figure that unavoidable pressure destroyed exergy decreases slightly with the gradual increase of buried depth, but the proportion decreases, which causes unavoidable thermal destroyed exergy in the pipeline to basically remain consistent with the downward trend of unavoidable total destroyed exergy. This is because, with the gradual increase of buried depth, the total heat transfer coefficient of the pipe gradually decreases, which decreases unavoidable destroyed exergy. Additionally, the total destroyed exergy in the pipeline gradually decreases with increased insulation layer thickness, and the reduction trend slows. Therefore, to save energy and optimize waxy crude oil pipeline operations, it is necessary to consider the relationship between buried depth and destroyed exergy to determine the optimum buried depth for transportation.

### 4.4. Influence of Operational Parameters on Unavoidable Destroyed Exergy

The effects of operational parameters, such as throughput, outstation temperature and outstation pressure, on unavoidable destroyed exergy are as follows:

#### 4.4.1. Throughput

Waxy crude throughput is varied (50, 60, 70, 80, 90 and 100 m^3^/h). Destroyed exergy during pipe transportation is calculated and destroyed exergy distribution curves are shown in Figure 8.

From Figure 8, it can be seen that crude oil throughput rises, and unavoidable pressure destroyed exergy first decreases and then gradually increases, although the proportion decreases. This causes unavoidable thermal destroyed exergy in the pipeline to basically remain consistent with the unavoidable total destroyed exergy change trend. It increases significantly first, and then slows. This is related to heat dissipation during pipeline transportation. The heat transfer coefficient of crude oil in the Newtonian interval is the largest, the non-Newtonian turbulence interval is second, and the non-Newtonian laminar flow interval is the smallest. With the increase of throughput, the Newtonian interval lengthens during pipeline transportation, but the non-Newtonian interval shortens. When throughput increased from 50 m^3^/h to 60 m^3^/h, the non-Newtonian laminar interval disappears during pipeline transportation, and the Newtonian and non-Newtonian turbulence intervals lengthen. With the increase of the total heat transfer coefficient, a greater temperature difference is needed as a driving force to ensure the smooth transportation of crude oil, which increases unavoidable thermal destroyed exergy in the pipeline. The interval length and heat transfer coefficient of crude oil with different throughputs are shown in Table 5.

With the gradual increase of the throughput of crude oil, total destroyed exergy in the pipeline will rise correspondingly. The unavoidable total destroyed exergy first increases significantly and then the increasing trend slows. After the increase to 80 m^3^/h, the increase unavoidable destroyed exergy will not exceed 2%. This outcome indicates that there is an optimal throughput in waxy crude oil pipeline transportation. When the pipeline exceeds this throughput, the avoidable destroyed exergy of the pipeline can rise, and a large quantity can be lost to the environment. Therefore, the confirmation of optimal crude oil throughput in an oil pipeline is very important to reduce the destroyed exergy of pipeline transportation and ensure the smooth operation of pipeline transportation.

#### 4.4.2. Outstation Temperature

Waxy crude outstation temperature is varied (55, 60, 65, 70 and 75°C). Destroyed exergy during pipe transportation is calculated and destroyed exergy distribution curves are shown in Figure 9.

It can be seen in the figure that crude oil outstation temperature will first increase and then gradually decrease, and its proportion decreases, which causes unavoidable thermal destroyed exergy in the pipeline to basically remain consistent with the unavoidable total destroyed exergy change trend (which increases significantly first, and then slows). After the increase to 65 °C, the increase of unavoidable destroyed exergy will not exceed 4%. The total destroyed exergy in the pipeline transportation process increases with increased outstation temperature, as does the avoidable destroyed exergy. Therefore, low temperature operation can save energy in crude oil pipeline transportation. However, due to the wax crude oil in China, it is easy to produce condensate pipe accidents at low temperatures. Thus, it is necessary to determine the optimal crude oil outstation temperature to ensure the safe and economic operation of crude oil pipelines. 

#### 4.4.3. Outstation Pressure

Waxy crude outstation pressure is varied (4000, 4250, 4500 and 5000 kPa). Destroyed exergy during pipe transportation is calculated and destroyed exergy distribution curves are shown in Figure 10.

From the distributions in the figure, unavoidable pressure destroyed exergy decreases slightly with the rise of outstation pressure, but the proportion decreases, which causes unavoidable thermal destroyed exergy in the pipeline transportation process to basically remain consistent with the unavoidable total destroyed exergy change trend. There is a slight decrease, but it has a very small effect. Additionally, the total destroyed exergy of the pipeline increases slightly with rising outstation pressure because the change of outstation pressure has little effect on the temperature drop along the pipeline, and so, the change of unavoidable thermal destroyed exergy is very small. In addition, the change of crude oil viscosity is relatively small, which ensures that the unavoidable destroyed exergy changes less for oil safety and economic transmission. Therefore, it is necessary to further consider the relationship between outstation pressure and destroyed exergy to ensure optimal energy conservation operation by determining the optimum outstation pressure of the wax crude oil pipeline.

## 5. Conclusions

(1) Irreversible phenomena, such as heat transfer and flow in a crude oil pipeline, inevitably lead to destroyed exergy. However, conventional exergy analysis is based on the ideal process of non-drive power and can suggest possible process improvements, but cannot determine whether the improvement is feasible. Based on the minimum theoretical temperature and pressure drops required in the crude oil safe transportation, unavoidable and avoidable destroyed exergy are defined. Energy savings will be changed from a general reduction of destroyed exergy to decrease the unavoidable destroyed exergy as far as possible.

(2) Wax crystallization can transform crude oil from a Newtonian fluid to non-Newtonian fluid. Therefore, to consider the influence of wax precipitation on unavoidable destroyed exergy during pipe transmission, it is necessary to divide the pipeline into intervals according to the wax precipitation point, anomalous point and flow transformation temperature. The formulas for unavoidable destroyed exergy of oil products under different flow conditions are derived, which comprehensively consider the flow characteristics of oil products in those transportation intervals. Compared with the simple assumption that the oil in the pipeline is in a single Newtonian flow state, this method can reflect the actual loss of the pipeline. 

(3) The applied example shows unavoidable thermal destroyed exergy in waxy crude oil pipeline transportation accounts for the majority of the unavoidable destroyed exergy, reaching 95–99%. The maximum and minimum unavoidable destroyed exergy in different intervals are 381.128 kJ/s and 30.259 kJ/s. The unavoidable destroyed exergy of the whole pipeline 742.934 kJ/s. It is necessary to further analyze the variation of unavoidable destroyed exergy with different design and operational parameters during pipeline transportation. The research shows that with the rise of the oil pipeline diameter, unavoidable destroyed exergy will first increase up to 625 kJ/s, then gradually decrease, and the overall variation will not exceed 15%. With the increase of insulation layer thickness, unavoidable destroyed exergy will decrease. The unavoidable pressure destroyed exergy is least at 22 kJ/s in 30 mm. Unavoidable destroyed exergy decreases slightly with the lower burial depths. Unavoidable destroyed exergy will settle in 610 kJ/s, but the increase amplitude will not exceed 2% after crude oil throughput arrives at 80 m^3^/h. With the rise of outstation temperature, unavoidable destroyed exergy first increases significantly, then the increase trend slows, especially when the outstation temperature rises to above 65 °C; the increase of unavoidable destroyed exergy will not be more than 4% at 618 kJ/s. With the rise of outstation pressure, unavoidable destroyed exergy will increase slightly. The above research results show that it is necessary to determine the optimal parameters for waxy hot oil pipeline transportation for energy conservation.

Through theoretical and case study, we further deepen the analysis. The unavoidable destroyed exergy calculation can find out the weak links of energy. And it can be adjusted by changing the different working conditions to achieve the most effective results. For example. In the practical engineering application, it is necessary to keep the high flow and the outstation pressure stable, enlarge the diameter as far as possible, increase the thickness of insulation layer and raise the outstation temperature within the scope of meeting the relevant standards. It plays an important role in the research of waxy crude oil pipeline.

## Figures and Tables

**Figure 1 entropy-21-00058-f001:**
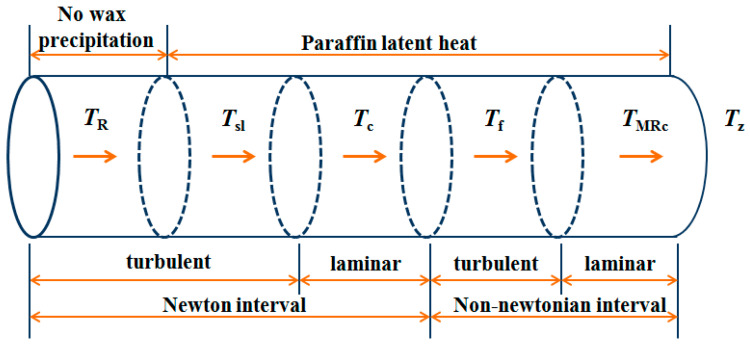
The division of the temperature range for a waxy crude oil pipeline.

**Figure 2 entropy-21-00058-f002:**
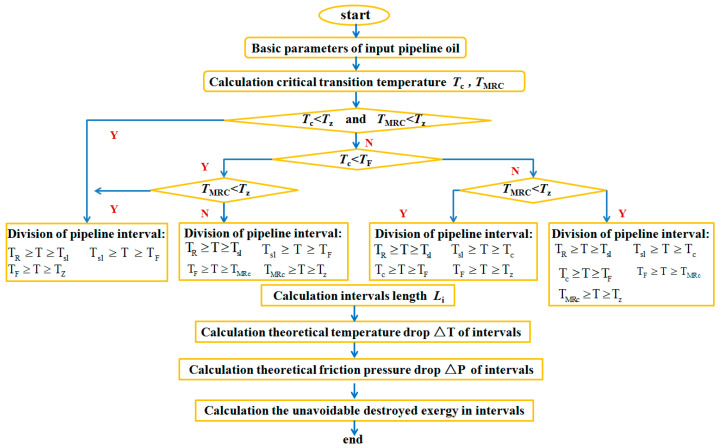
Unavoidable destroyed exergy calculation flow chart for crude oil pipeline transportation process.

**Figure 3 entropy-21-00058-f003:**
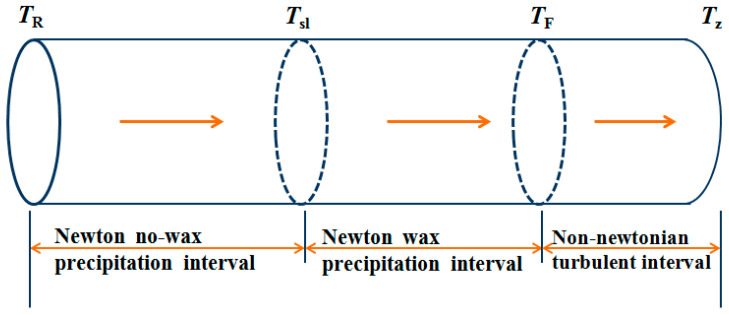
Temperature regions of a waxy crude oil.

**Figure 4 entropy-21-00058-f004:**
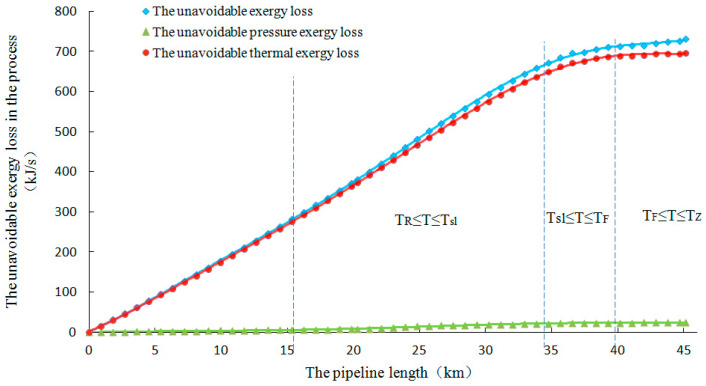
Unavoidable destroyed exergy distribution curve of the waxy crude oil pipeline

**Figure 5 entropy-21-00058-f005:**
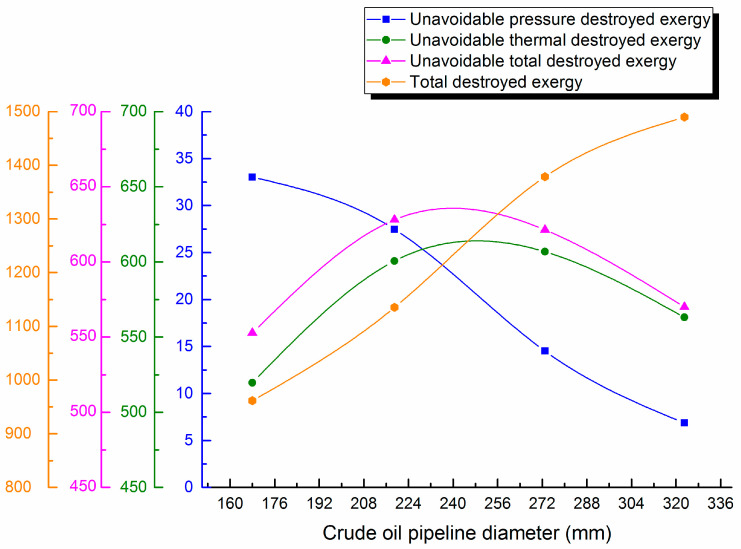
Unavoidable pressure destroyed exergy, unavoidable thermal destroyed exergy, unavoidable total destroyed exergy and total destroyed exergy against diameter.

**Figure 6 entropy-21-00058-f006:**
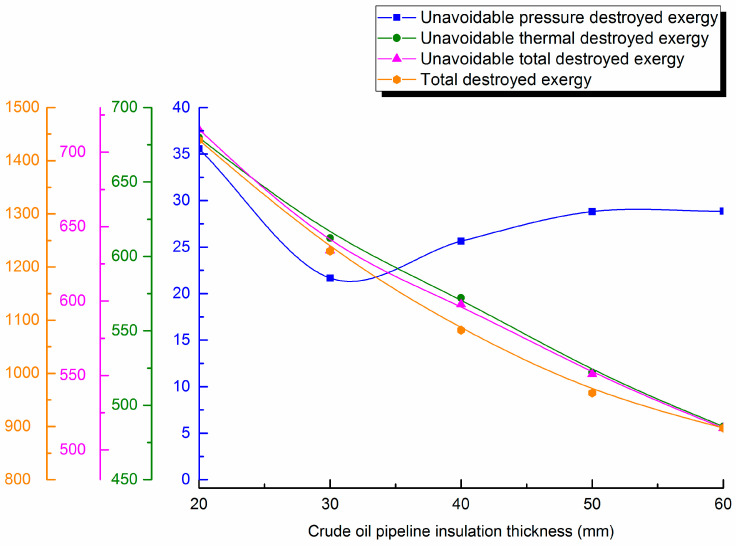
Unavoidable pressure destroyed exergy, unavoidable thermal destroyed exergy, unavoidable total destroyed exergy and total destroyed exergy against insulation thickness.

**Figure 7 entropy-21-00058-f007:**
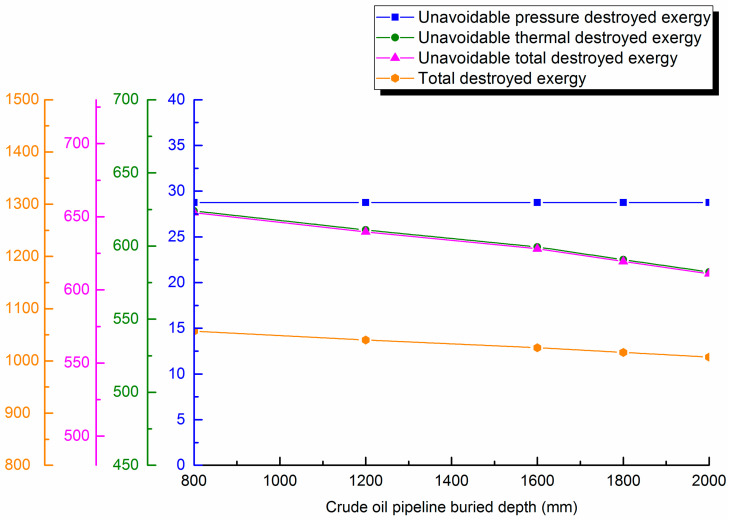
Unavoidable pressure destroyed exergy, unavoidable thermal destroyed exergy, unavoidable total destroyed exergy and total destroyed exergy against buried depth.

**Figure 8 entropy-21-00058-f008:**
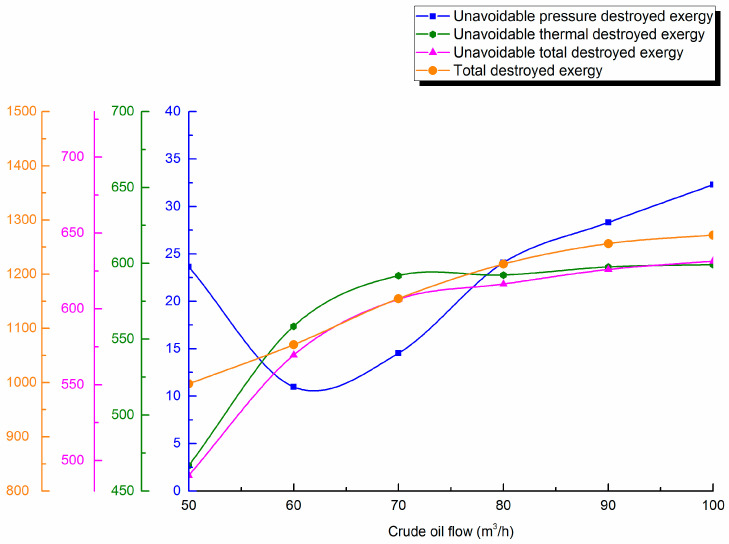
Unavoidable pressure destroyed exergy, unavoidable thermal destroyed exergy, unavoidable total destroyed exergy and total destroyed exergy against throughput.

**Figure 9 entropy-21-00058-f009:**
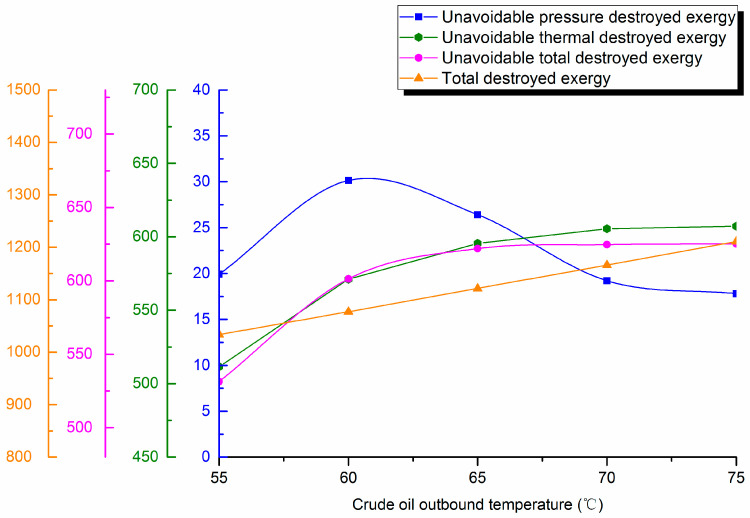
Unavoidable pressure destroyed exergy, unavoidable thermal destroyed exergy, unavoidable total destroyed exergy and total destroyed exergy against outstation temperature.

**Figure 10 entropy-21-00058-f010:**
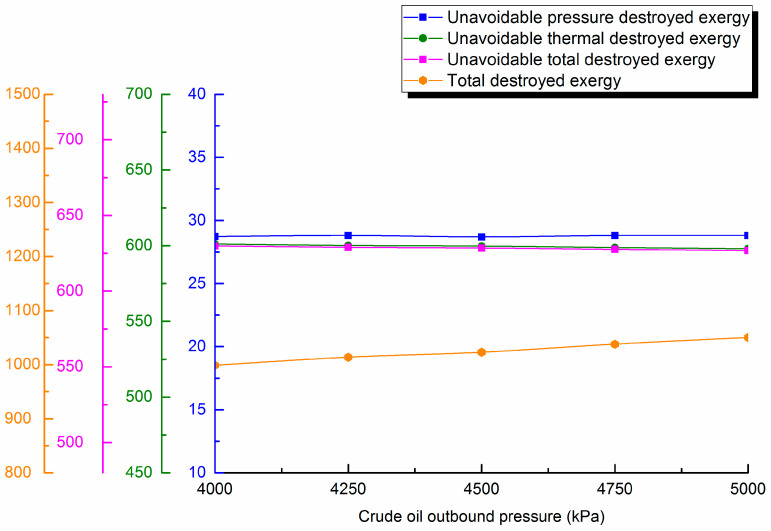
Unavoidable pressure destroyed exergy, unavoidable thermal destroyed exergy, unavoidable total destroyed exergy and total destroyed exergy against outstation pressure.

**Table 1 entropy-21-00058-t001:** The parameters for four kinds of crude oil [19].

Parameter	A	B	n	m
Shengli crude oil	0.4840	1.9255	0.03465	0.01164
Daqing crude oil	0.9085	1.7585	0.01732	0.01567
Puyang crude oil	0.6753	1.7258	0.0264	0.01217
Renqiu crude oil	0.1970	1.8880	0.0476	0.02117

**Table 2 entropy-21-00058-t002:** Basic operating parameters of a Daqing oil pipeline.

Item	Data	Item	Data
Pipeline length	45121 m	Pipeline diameter	Φ219 × 5.6 mm
Buried depth	1600 mm	Wax Appearance point	47.7 °C
Outstation temperature	65 °C	Anomalous point	36.2 °C
Outstation pressure	4.5 MPa	Condensation point	25 °C
Pipeline throughput	70 m^3^/h	Density (30 °C)	860 kg/m^3^
Ambient temperature	−4.4 °C	Density (50 °C)	830 kg/m^3^
Soil thermal conductivity	1.4 W/(m·°C)	Viscosity (30 °C)	70 mPa·s
Pipe thermal conductivity	45.24 W/(m·°C)	Viscosity (50 °C)	9.41 mPa·s

**Table 3 entropy-21-00058-t003:** The calculation results of unavoidable destroyed exergy in the waxy crude oil pipeline transportation process.

Pipeline Transmission Temperature Region	Newton Turbulent Non-Wax Precipitation	Newton Turbulent Wax Precipitation	Non-Newton Turbulence	Whole Process of Crude Oil Pipeline Transportation
	$T_{R}\geq T\geq T_{sl}$	$T_{sl}\geq T\geq T_{F}$	$T_{F}\geq T\geq T_{Z}$	$T_{R}\geq T\geq T_{z}$
The unavoidable pressure destroyed exergy (kJ/s)	7.749	15.828	0.042	23.621
The unavoidable thermal destroyed exergy (kJ/s)	373.379	315.716	30.217	719.313
The unavoidable destroyed exergy (kJ/s)	381.128	331.544	30.259	742.934
The ratio between unavoidable pressure destroyed exergy and pressure destroyed exergy (%)	9.409	22.202	0.328	14.178
The ratio between unavoidable thermal destroyed exergy and thermal destroyed exergy (%)	58.429	61.883	34.786	58.193

**Table 4 entropy-21-00058-t004:** Crude oil pipeline temperature intervals under different pipe diameters.

Pipeline Diameters		168 mm	219 mm	273 mm	323 mm
Interval length of pipe transmission (km)	Newton no-wax precipitation	37.450	28.877	24.816	21.685
	Newton wax precipitation	7.671	16.244	20.305	21.631
	non-Newton turbulence	/	/	/	1.805
End point temperature (°C)	Newton no-wax precipitation	43.83	47.81	47.64	47.86
	Newton wax precipitation	40.47	40.26	37.39	36.61
	non-Newton turbulence	/	/	/	35.85
Theoretical outstation temperature (°C)		41.32	43.44	43.64	42.57

**Table 5 entropy-21-00058-t005:** Calculation results of crude oil pipeline temperature intervals under different throughputs.

Throughput		50 m^3^/h	60 m^3^/h	70 m^3^/h	80 m^3^/h	90 m^3^/h	100 m^3^/h
Pipeline interval length (km)	Newton no-wax precipitation	14.438	17.597	20.304	23.914	27.975	29.779
	Newton wax precipitation	14.438	16.243	18.951	21.207	17.146	15.342
	Non-Newton turbulence wax precipitation	8.573	11.281	5.866	/	/	/
	Non-Newton laminar wax precipitation	7.671	/	/	/	/	/
Total heat transfer coefficient (W/m^2^ °C)	Newton no-wax precipitation	0.469	0.472	0.470	0.476	0.471	0.471
	Newton wax precipitation	0.447	0.460	0.468	0.467	0.469	0.470
	Non-Newton turbulence wax precipitation	0.416	0.42652	0.461	/	/	/
	Non-Newton laminar wax precipitation	0.356	/	/	/	/	/

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
