# Peer review of "Unavoidable Destroyed Exergy in Crude Oil Pipelines due to Wax Precipitation"

_entropy, 2019, doi:10.3390/e21010058_

Round 1
Reviewer 1 Report
The authors analyzed the exergy loss in crude oil pipelines in much detailed description of flow regimes and temperature regimes. The analysis in this paper seems to be helpful in crude oil transportation. However, the descriptions/narratives are hard to understand. Clarifications and rewording are necessary for better conveying the findings to readers. The decision will be reconsidered after revision.
In Section 3.1, it seems like that non-Newtonian behavior and the turbulence happen together above Tc. Is it correct or misleading? It is recommended why non-Newtonian behavior happens and why turbulence happens at the same or similar condition.
In Section 3.3, Tq is not defined ..not even in the Nomenclature. That makes the rest of the part hard to understand.
Isn't it better to also plot the curve of avoidable total exergy loss?
Isn't it just better to minimize the total exergy loss? It is not clear what kind of optimization is required. for example of fig 5, isn't it just better with lesser exergy loss with smaller diameter. The result does not show what kind of trade off must be considered in the optimization. The rest of the results have the same issue.
very minor thing. please switch the color of total exergy loss and unavoidable total exergy loss in figure 8 for the consistency with other figures.
Additionally, are there any real examples or cases to support the results? The analysis seems to be theoretical feasibility study. I think that is the editor's decision whether that is enough for publication in this journal or not.
Author Response
Thank you for your careful review. We have revised it as required and look forward to your reply.

Reviewer 2 Report
I do not understand this paper, and I doubt that most other readers would as well. I have heard of exergy, but it is not a property that most engineers are familiar with. Do not assume that your readers know what exergy is, or that they believe that it is relevant to oil transport. Give a precise thermodynamic definition of exergy. Try to convince the reader that exergy is important to oil transport.
Author Response
Thank you for your careful review. We have revised it as required and look forward to your reply

Reviewer 3 Report
The most important merit of the paper is that the authors improved the methodology of the exergy loss analysis of crude oil pipelines. Instead of using a single, “average flow regime, they divide the total length into section where different flow regimes exist. To do that, they use the temperature dependence of the rheological behavior (constitutive laws) to determine the characteristic temperatures where the crude oil changes from Newtonian to non-Newtonian fluid and the flow transits from laminar to turbulent. Their methodology systematically uses the splitting of the exergy losses into unavoidable and avoidable parts.
It is also important that the authors performed a parametric study to analyze the influence of design (diameter, insulation, buried depth) and operational parameters (flow rate, outlet temperature and pressure) on the thermal and pressure related exergy losses.
The weak point of the manuscript is its form: style, language and lack of editing.
A few examples of sloppy formulations, usage of undefined or unusual terminology:
41 - “…to determine the gapsbetween various systems…”
56 - “…effectiveexergy consumption and invalidexergy consumption…”
100-101 -it would help the reader if the authors define clearly the characteristics temperatures as “wax appearance point”, “flow transition temperature”, “anomalous point”; the authors do not consequently these names, there is also “critical transition temperature”
112-122 – The internal diameter of the pipeline is denoted by “D” as well as “d” throughout the manuscript
- The length of the pipeline is Lin (12) vs lin the Nomenclature and in (19); its unit is said to be km, but the formulae suggest that it is in meter.
- “I” is stated as “hydraulic gradient” in 190, but as “oil flow gradient”in the Nomenclature
119 - “…the specificationprovides that the critical Reynolds number is 2000…” – what specification ?
129 - “counter-calculated” – “back calculated”
150 - a brief explanation of the existence of a maximal wax thickness (“peak”) would help many readers
152 - decreased -> decreasing
157 - “The wax deposition tail section spans the peak of wax deposition to the interval of crude oil intake.” - The sentence is not clearly formulated.
160 - “…the specific heat capacity of crude oil decreases with reductionoil temperature.” -> “…the specific heat capacity of crude oil decreases with decreasingoil temperature.”
192 - unusual terminology: “microelement” -> “elementary”
196 - The Lilengths do not come from the formula (9)
242 - in equation (22) the sign of prime in n’and the exponents are fused together
245 - the formula (22) is not an integral; Ex,Ui is not temperature range, even if the temperature range can be back-calculated from it, as it was said earlier
255 - Subtitle 3.5: “The process of…exergy loss” or rather the “Process of calculating of the …exergy loss”?
266 - in Table 2 the authors show the thermal conductivity of the soil and that of the pipe – without explanation
373 - paragraph 3); there is no any hint about the method of calculating the heat losses of a pipeline buried in the soil. When thermal exergy losses were mentioned earlier, an overall heat transfer coefficient K figured in the formulae. No any mention of a conduction resistance or shape factor is mentioned that could help illuminating the method used for the calculation of the effect of the depth…
453 - The reviewer suggests a stylistic revision of the Conclusions. Better structure, better distinction of the different findings which are fused together in the present text; revision of the English formulations.
Terminology, capitalization:
outbound – outlet (pressure, temperature) – exit - terminal etc.
newtonian <-> non-Newtonian
Author Response

(The authors gave the same response as above.)

Reviewer 4 Report
The work presents an interesting and innovative approach for the problem of wax precipitation, which is an important issue for the petroleum industry. However, some aspects of the paper should be revised.
- First of all, the authors must emphasize what they mean by “exergy loss”. There are different nomenclatures used in the literature, but, most of the times, the term exergy loss refers to an amount of exergy that is neither destroyed nor used by the system/control volume to perform work; it is released to the environment instead. For instance, it can refer to the exergy transfer to the environment associated to heat transfer. When talking about the reduction of the exergy due to irreversibilities, the term destroyed exergy is more accepted. The authors can keep “exergy loss”, but they must highlight, at the beginning of the paper, what they mean by that.
· The paragraph that begins in line 49 must be rewritten in order to provide more clarity for future readers. Moreover, from line 69, the authors present a short discussion about exergy analysis that contains some misconceptions. They say that the exergy analysis takes into account ideal processes. That is not correct. The definition of exergy is, indeed, the maximum amount of work that could be obtained by taking a system/flow to a state of equilibrium with the environment by means of reversible processes. However, the exergy analysis is applied to real processes, not only reversible ones. That’s why we have in the exergy balance the parcel of the exergy that is destroyed in the process. This term shows how far the real process is from the reversible one. The authors should spend more time understanding the exergy method in order to correct some aspects of the introduction.
· In line 86 the authors attribute to Zhang Xiaohui et al. the development of the idea of unavoidable exergy loss. However, as far as I know, this concept was first brought up by Tsatsaronis and Park in 2002 (On avoidable and unavoidable exergy destructions and investment costs in thermal systems). This should be corrected in the article.
· Moreover, most of the references presented by the authors are from Chinese researchers. When talking about petroleum, I am sure that other countries, such as the US and Norway, also have plenty of literature worth reading about the subject. As far as exergy analysis is concerned, researchers from Germany, Italy, Spain and Brazil have a lot of material useful for the work being developed by the authors. A comprehensive literature review, which includes works from research groups from different countries, gives more credibility to the paper.
· In line 126, “for Newtonian waxy crude oil, the critical Reynolds number is 2000”. 2000 is critical Reynolds number for internal flow.
· Equation 6 was taken from the literature. The authors should check if it is dimensionless.
· The K’ is presented in line 140 with unit Pa.sn’, but is determined by equation 7, where there is no exponent n’. Actually, all equations are taken from the same reference and are lacking further explanations, specially about their origin, if they are analytical or empirical, if the constants were obtained by means of linear/polynomial regression. The same goes for the data in table 1. They also come from reference 10, but were they obtained experimentally? That kind of information must be provided
· Equation 12 presents the energy balance of the process, also from reference 10. The authors must inform what each parcel of the balance means (heat transfer due to conduction, convection, heat “generation”, etc). Moreover, some variables are not detailed, neither in the text nor in the nomenclature.
· In Equation 13, “D” is replaced by “d”.
· The formulation presented in Equation 16 for the exergy destruction is valid for incompressible fluid with constant specific heat in an adiabatic process. It should be mentioned.
· For the sake of standardization, it would be interesting to change the variable “h” in equation 19 for another letter, so the equation is not mistaken by the expression for the hydrostatic pressure. And what is the variable β?
· It was not clear to me if the reduction of the internal diameter due to wax deposition was taken into account.
· In the sensitivity analysis presented by the authors, the unavoidable pressure exergy loss shows a different behavior in comparison to the other ones. The authors associate it with the heat transfer only, but since it is the pressure contribution, other aspects should be further explored.
· In addition, there are many graphs showing the behavior of the variables, but the most important topic about avoidable and unavoidable exergy destruction is brought up only in the last paragraph of the conclusions, which is what practical points can be addressed in order to reduce the avoidable exergy destruction. The author generated a good amount of information, but they did not explore it properly. The results should be better discussed.
Author Response
We have made changes and resubmitted them

Round 2
Reviewer 1 Report
Now it is much improved than the previous version.
Now please consider the following styles for better readability.
The way of citing other papers. To the best of my knowledge, only the last name is used. For example, line 44..Zuo Lili.... not that way but..only Zuo.... additionally, the line 40 sounds awkward to international readers... Chinese scholar .... you don't have to say that. Anyone who publishes paper is a scholar and anyone can search that name if they are really interested in where he/she came from.
Why the variables in the sentence has a weird position? for example in line 92, Ex,U is placed a little above the text line. Collaborate with the editorial office to fix that
Figure 2 must be improved. It is hard to read...especially all the temperature regimes. Please make each font larger.
Author Response

(The authors gave the same response as above.)
